# Working Conditions, Musculoskeletal Pain and Wellbeing Among Hospital Surgeons: A Cross-Sectional Study

**DOI:** 10.3390/healthcare13080898

**Published:** 2025-04-14

**Authors:** Georgia Ntani, Stefania D’Angelo, Robert Slight, Lesley Kay, Michael Whitmore, Dan Wood, Karen Walker-Bone

**Affiliations:** 1MRC Lifecourse Epidemiology Centre, University of Southampton, Southampton SO16 6YD, UK; gn@mrc.soton.ac.uk (G.N.); sd@mrc.soton.ac.uk (S.D.); 2MRC Versus Arthritis Centre for Musculoskeletal Health and Work, MRC Lifecourse Epidemiology Centre, University of Southampton, Southampton SO16 6YD, UK; 3Population Health Sciences Institute, Newcastle University, Newcastle upon Tyne NE2 4AX, UK; bob.slight@newcastle.ac.uk; 4The Newcastle upon Tyne Hospitals NHS Foundation Trust, Newcastle upon Tyne NE7 7DN, UK; 5Rheumatology Department, The Newcastle upon Tyne Hospitals NHS Foundation Trust, Newcastle upon Tyne NE7 7DN, UK; bendangus@googlemail.com; 6RAND Europe, Cambridge CB2 8BF, UK; mwhitmor@randeurope.org; 7Urology, University of Colorado, Aurora, CO 80045, USA; isovialdan@gmail.com; 8Department of Urology, University College London Hospitals (UCLH), London W1G 8PH, UK; 9Monash Centre for Occupational and Environmental Health, Monash University, Melbourne 3004, Australia

**Keywords:** surgeons’ health, healthcare workers, pain, wellbeing, work–life conflict, retirement

## Abstract

**Background/Objectives:** Patient safety is directly linked with health and wellbeing of healthcare workers. In the UK, COVID-19 severely disrupted healthcare, with surgeons tackling prolonged waiting lists and working longer hours under high stress. This study explored the biomechanical and psychosocial demands on hospital surgeons, as well as their experience of pain and work-life balance post-pandemic. **Methods:** A questionnaire was developed combining validated tools assessing physical demands; modified job demand, control, and support; the WHO-5 wellbeing index; work–life balance from the Copenhagen Psychosocial Questionnaire; musculoskeletal pain; job satisfaction and retirement intentions. An online survey was developed using the Qualtrics© (Provo, UT, USA) platform and circulated through surgical networks using snowball sampling. Poisson regression modelling with robust confidence intervals was used to explore relationships between work-related factors and musculoskeletal pain, and associations with retirement intentions. **Results:** In total, 242 replies were received. Surgeons frequently reported strenuous occupational activities and work–life imbalance, and one in six reported job dissatisfaction. Only 17% reported no pain; the one-month prevalence of pain ranged from 46% at the lower back to 12% at the ankle, and pain was frequently disruptive. Better work–life balance had a protective effect for pain (PRR = 0.92, 95% CI = 0.85–0.99), while risk of pain increased with increasingly physically demanding activities at work (PRR = 1.04, 95% CI = 1.01–1.07) in the age- and sex-adjusted models. Job dissatisfaction was associated with intention to retire early (PRR = 1.83, 95% CI = 1.02–3.27). **Conclusions:** This study demonstrated high physical and mental demands among surgeons and poor work–life balance. Physical and emotional links to pain were identified. Fit surgeons ensure safe patient care. Our findings suggest that surgeons were facing health issues and work–life conflict post-pandemic, potentially limiting their job satisfaction and career span. A follow-up study is recommended.

## 1. Introduction

There is substantial evidence that poor wellbeing of healthcare workers is associated with poorer patient safety [1]. The coronavirus SARS-CoV-2 (COVID-19) pandemic substantially disrupted global healthcare systems in 2020–21. Many hospitals closed to elective activity and acutely ill respiratory patients were cared for by all staff, including surgeons. Concurrently, lockdown restrictions significantly reduced primary care consultations [2], which markedly impacted healthcare-seeking and uptake of screening and routine follow-up care [3,4,5]. In the UK, 72% of elective operations were cancelled within 12 weeks of the planned date [6]. Understandably, surgical referrals increased dramatically when the pandemic was brought more effectively under control.

Since then, healthcare systems have attempted to recover from the overwhelming burden of the backlog in services and the longer-term consequences of the pandemic [7]. In England, although the backlog is reducing, there are still an estimated 6.24 million patients on the waiting list for NHS care in January 2025, despite record numbers of patients treated in 2024 [8]. Beyond the profound consequences for patients, this backlog of cancelled appointments and treatments means that the services, and the staff delivering them, remain under considerable strain to triage and deliver effective patient care.

Surgeons were at the forefront of restoring patient care and addressing the increased waiting lists. Amongst working-age adults, the leading causes of sickness absence, presenteeism and work disability are musculoskeletal disorders and common mental health conditions [9]. Globally, healthcare struggles to retain its workforce, particularly doctors and nurses [10]. From 2007 to 2020, the number of doctors working in the NHS opting for early retirement trebled [11], impacting patient care, health service delivery and surgical training [12]. Even pre-pandemic, there was evidence that surgeons were commonly reporting musculoskeletal disorders [13,14,15,16] with conflicting evidence as to whether pain was increased or decreased by robotic/video-assisted techniques [17,18]. Thus, the aim of this study was to obtain a post-pandemic understanding of surgeons’ working conditions, perceived work–life conflict, psychological wellbeing, retirement intentions and experiences of musculoskeletal pain in order to facilitate workforce planning and maximise the efficiency of surgical practice while fostering a healthier and more resilient healthcare workforce.

## 2. Material and Methods

### 2.1. Participants and Procedure

A brief questionnaire entitled “The health, work and wellbeing of Surgeons” was developed on the Qualtrics© on-line platform. The survey was publicised through co-authors’ surgical networks predominantly in the UK but also in the USA using societies and social media and snowball sampling between October 2021 and January 2022.

### 2.2. Outcomes

#### 2.2.1. Musculoskeletal Pain

Musculoskeletal pain was assessed through mannequins illustrating eight anatomical regions (lower back, hip, neck, shoulder, elbow, wrist/hand, knee, and ankle/foot). For each site, participants reported pain lasting for a day or longer in the past month, and its impact on normal and/or recreational physical activities. They also reported if the pain had led them to seek health professional advice/treatment, and/or take analgesia. The outcomes were ‘any pain’, defined as pain occurring anywhere in the body in the past month, and ‘any disabling pain’, defined as any pain occurring in the past month that interfered with normal activities.

#### 2.2.2. Early Retirement

Participants reported if they intended to practice until they became eligible for state pension (currently aged 67 years in the UK). They were classed as intending to retire earlier than their state pension age if they replied negatively and were compared with the rest of the sample.

### 2.3. Covariates

Relevant covariates were chosen to be brief and cover each of the following specific domains: psychological wellbeing; ergonomic exposures; psychosocial exposures; work–life conflict; job satisfaction and working conditions.

Psychological wellbeing was measured using the World Health Organisation Five Well-Being Index (WHO-5), assessing a two-week span through five questions, each with six answer options. Scores ranged from 0 to 5, summing up to an overall percentage score (0–100). Higher values represented better overall wellbeing. A cut-point of 50 dichotomised the overall wellbeing score into low (≤50) and better (>50) wellbeing [19].

Surgeons reported on an average working day their exposure to the following: standing (most or >3 h/day); sitting (most of the day); working with the neck twisted (>1/2 h/day); working with neck flexed (>2 h/day); using keyboard (>1 h/day and >4 h/day); repeating movements with wrist/fingers (>1 h/day and >4 h/day); bending/straightening the elbow (>1 h/day); using hand-held vibrating tools (>1 h/day); and working with hand/wrist rotated (>1 h/day). They estimated their weekly duration operating and the proportion of their operating time spent standing/sitting.

Work-related demand, control and support was assessed using questions from the validated Swedish Demand-Control-Support Questionnaire (DCSQ) [20], the modified version of Karasek’s questionnaire [21]. A total of 11 questions were used, each with four answer options from ‘Never’ to ‘Frequently’. Higher scores indicate higher perceived demands and perceived control, and a combination generated DCSQ work categories: Low strain (Low demands/High control); Active job (High demands/High control); Passive job (Low demands/Low control); and High strain (High demands/Low control). Low strain was used as the reference category.

To evaluate work–life conflict, we used questions from the third version of the Copenhagen Psychosocial Questionnaire (COPSOQ III) [22], with response options from ‘always’ to ‘never/hardly ever’, scored from 1 to 5, respectively. To create a work–life balance index, we averaged responses across the five questions, with higher scores on the index indicating better work–life balance. Participants with a missing value in three or more questions were assigned a missing value in the work–life balance index.

Job dissatisfaction was assessed with a single question. Response categories were ‘very dissatisfied’, ‘dissatisfied’, ‘satisfied’ and ‘very satisfied’, with the first two and the last two grouped together to generate a binary variable ‘dissatisfied versus not’, as has been done previously [23].

The following questions were asked about working conditions: (1) “I can find a chair suitable for my needs at work”, (2) “I can find a lead apron suitable for my needs when required”, (3) “I can adjust the operating table in theatre for comfort”, (4) “I try to give priority to achieving the best posture for my comfort when at work”, and (5) “I take planned breaks whilst at work”. Participants were then classed as having unfavourable working conditions if they replied ‘rarely’/‘never’ (as opposed to ‘frequently’/‘some of the time’) to at least three of the five statements.

Physical comfort provision at work was assessed by asking whether participants received any formal or informal assistance or training on ergonomics and posture. Those answering negatively were classed as lacking provision for physical comfort at work.

Leisure time physical activity data included participants’ engagement in specified physical activities on an average workday. The International Physical Activity Questionnaire Short Form (IPAQ-SF) was used to assess weekly frequency and time engaged to vigorous, moderate physical activity and walking, later categorised into low, moderate and high physical activity according to guidelines for data processing of IPAQ-SF [24].

### 2.4. Statistical Analysis

We used counts and percentages to describe the study population, their characteristics, and musculoskeletal pain. We used Poisson regression models with robust standard errors to explore the effects of personal factors, demographic factors, and work-related physical and mental influences on musculoskeletal pain, and also on intention to retire before state pension age. The effects were summarized by prevalence rate ratios (PRRs) and 95% confidence intervals (CIs) and were sex- and age-adjusted. A final mutually adjusted model was also computed with estimates adjusted for age, sex and all factors significant in the minimally adjusted models.

Data were analysed using Stata v17.0 software.

## 3. Results

In total, 242 surgeons, aged 25–65+ years, completed the survey. A total of 70% were males and one third were aged 45–55 years (Table 1). Respondents were from a range of sub-specialities, with the highest response rate among urologists (66%). Almost half were very physically active. One third reported low levels of psychological wellbeing (35%).

Surgeons reported operating for a median of 12 h/week, with 65% standing and 35% sitting. Open surgery was done by 25%, mostly in abdominal–pelvic cases. Imaging-guided surgery was performed ~10% of the time, laparoscopic surgery <25% and robotic surgery <10%. Many surgeons reported physically demanding activities, including standing for >3 h (76%), working in awkward postures (>62%), repetitive wrist/finger movements (62%), and repetitive bending/straightening of the elbow (48%). Overall, participants were exposed to a median of 5/8 of the listed physically demanding activities. About 60% reported low-demand with low-control jobs, 29% high-demand with low-autonomy (15). A total of 64% lacked physical comfort provision, and 10% worked in unfavourable conditions. Overall, 17% of the surgeons were dissatisfied with their job (Table 2).

Figure 1 shows the impact of work on personal lives. The majority, at least often, reported completing work-related tasks during non-working hours (>74%), and being contacted outside working hours in emergencies (40%). Almost half reported that always/often work negatively impacted their private life (46%), that they changed plans due to work-related duties (47%), and that work demands interfered with their private life (48%). A quarter often juggled being at work and at home simultaneously (26%). Work–life imbalance score averaged 2.7 (SD = 0.8).

Musculoskeletal prevalence rates were high in most body sites (Table 3). The one-month pain prevalence ranged from 46% at the lower back to 12% at the ankle. Pain was also prevalent at shoulder (27%) and hip (23%). A total of 83% reported pain in at least one anatomical site. High rates of disability were seen in sites of low and high pain prevalence. For example, 40% of those with lower back pain reported related disability, similar to ankle/foot pain. Over 35% reported disability with hip, neck, and shoulder pain. Rates of interference with recreational physical activities were higher than those of interference with normal activities. More than half of those with hip, shoulder, or knee pain reported interference with recreational physical activities. Seeking health professional advice/treatment varied from 8% to 17%, yet use of pain relief medication rates were considerably higher (range: 24–54%) (Table 3).

Table 4 summarises associations of musculoskeletal pain with risk factors. Prevalence of disabling pain was higher among females and younger surgeons. After adjustment for age and sex, better work–life balance correlated with lower pain prevalence (PRR: 0.92) but lack of physical comfort provision at work also correlated with lower pain prevalence (PRR: 0.88). More physically demanding work was associated with increased prevalence of pain (PRR: 1.04). The association with physical work demands and lack of physical comfort provision remained robust in the mutually adjusted model. The association with work–life balance only remained significant at a 10% level.

Prevalence of disabling pain was higher among participants with low mood (PRR: 1.43), increased with increasing levels of physically demanding activity at work (PRR: 1.09), and decreased with better work–life balance (PRR: 0.63). Disabling pain increased with unfavourable working conditions, job dissatisfaction and high strain at work, but these associations were not statistically significant.

Among 176 surgeons, 24% reported intending to leave work earlier than state pension age. This intention was not associated with musculoskeletal pain (PRR = 1.88, 95% CI: 0.72–4.92) or related disability (PRR = 1.14, 95% CI: 0.68–1.92). Earlier retirement intention was, however, significantly associated with poorer work–life balance (PRR = 0.71, 95% CI: 0.51–1.00), and job dissatisfaction (PRR = 2.07, 95% CI: 1.23–3.49). In the mutually adjusted model, only the association with job dissatisfaction remained robust (PRR = 1.83, 95% CI: 1.02–3.27).

## 4. Discussion

This cross-sectional survey of 242 surgeons suggested physical and psychosocial challenges in their working environment and impacts of this on job satisfaction and work–life balance. One-third of respondents reported poor psychological wellbeing, and the majority reported high levels of physically demanding activities. Over half also reported that they had received no training or support for ergonomics/comfort from their employer. The one-week prevalence of pain was high, particularly at the lower back (46%), neck (44%) and shoulder (27%), and the pain was associated with disability for activities (>36% for each of the most prevalent pain sites). Poor job satisfaction was associated with an increased risk of reporting an intention to retire early from hospital work.

One-third of surgeons reported poor psychological wellbeing, consistent with the emotional distress (38%) and depression (29%) reported among arthroplasty surgeons in Canada during the pandemic [25], who attributed their distress to concerns about loss of income/operating time, emotional conflicts and worries about safety [26]. It is impossible to compare the reasons for poor wellbeing in the current survey as these questions were not asked. Much of the mental health research in surgeons pre-pandemic has focused on burnout, defined as a state of emotional weariness, depersonalisation and sense of personal failure that ultimately affects work performance [27]. Burnout leads to poorer patient safety (1), increased risks of errors [28], malpractice litigation, loss of temper and reduced empathy. Surgeons in a qualitative study perceived that burnout had impaired their communication and relationships, increasing the risk of making errors, and reducing their motivation to improve their practice [29]. Moreover, burnout and poor work–life balance are mutually associated [16]. Our study importantly demonstrated that surgeons were experiencing work–life conflict, evident in frequent work-related tasks outside of working hours, often disrupting private/family plans. It is clear that surgeons are vulnerable to psychological strain because of the nature of their job, and it is reasonable to expect that the pandemic may well have exacerbated this risk. Monitoring their mental health in the longer term is crucial to ensure that there is improvement, given the impact of this on the risk of medical and surgical errors.

Our study demonstrated surgeons’ high physical demands at work, including repetitive tasks and/or awkward postures, consistent with previous reports [30]. Despite this, surgeons reported only moderate levels of overall job dissatisfaction. Approximately one in six reported dissatisfaction, a rate consistent with other findings [9,31]. This is likely explained by the rewarding and fulfilling aspects of their job. Certainly, it has been shown in healthcare workers that “career calling” (passion or driving force motivating individuals in their work) positively impacts job satisfaction whilst job demands negatively impact it, whilst other job characteristics including autonomy, social support, feedback, management support and career opportunity can mitigate the negative effects of job demands on satisfaction [32].

High rates of musculoskeletal pain amongst surgeons have been reported before. A review of papers exploring work-related problems among UK surgeons found that musculoskeletal pain was particularly common in this population [12]. In a survey of surgeons at a tertiary care centre [33], 80% reported musculoskeletal complaints related to surgery, while a similar rate was also reported by Szeto et al. [34] amongst general surgeons in Hong Kong and in Iran [35]. In the current study, more than 80% reported pain in at least one anatomical site, while the prevalence of any disabling pain was 39%. The prevalence of disabling pain was higher amongst younger surgeons and female surgeons. It was noteworthy that whilst pain was prevalent and disabling, few surgeons had sought health professional advice/treatment (8–17%) but many had self-medicated (54%). Over half reported that they had never received any advice/support from their employer on ergonomic adjustments. It is possible that surgeons regard pain as part of normal working life that they need to just accept. With surgeons working under increased demands in 2021–22 due to the pandemic, it is perhaps not that surprising that they are experiencing high rates of pain associated with physical and psychological demands. However, it remains to be seen what the full impact of these symptoms could be, particularly if ignored, in the longer term.

Occupational physical activities were robustly associated with disability from musculoskeletal pain. High physical demands are unsurprisingly associated with pain as shown in a previous study of surgeons [36], while evidence for such an association is strong also in the wider working population [37,38]. This association could result from stresses and strains on tissues through maintaining abnormal postures or repetition, impacting the nervous system and pain perception resulting in symptoms [39].

Our findings of lower risk of disabling musculoskeletal pain for surgeons with better work–life balance aligns with previous reports. Studies in South Korea, Germany, Switzerland, and Boston found that work–life conflict was positively associated with musculoskeletal pain, independent of other work factors [40,41]. Work–life conflict, as the disparity between work and home demands, may induce distress which in turn can influence health, and particularly that affecting the musculoskeletal system, through its effect on muscular tension. This reinforces the biomechanical link between occupational psychological factors and musculoskeletal pain.

This study demonstrated that being dissatisfied at work and having a poorer work–life balance were associated with intention to leave before reaching state pension age. Job satisfaction has been previously shown to significantly affect intentions to leave work in many occupations, including healthcare workers [42], like nurses [43]. The risk of intention to retire earlier decreased with better work–life balance (PRR: 0.71), yet that association attenuated (PRR: 0.77) after adjustment for job dissatisfaction. Similar findings were reported in studies of Japanese nurses [44], and UK nurses and social care workers [45]. A study of physicians also found that work–family conflict was an important predictor of intention to leave [46]. Future research should explore whether work–life interference has a direct effect on people’s retirement intentions, or its effect acts by contributing to job dissatisfaction. Deeper understanding of the underlying mechanism will enable more effective policies for longer workforce retention both in the healthcare sector and beyond.

Our study has limitations: firstly, it is cross-sectional, limiting our ability to make causal inferences; secondly, we adopted a snowball sampling technique, hindering our ability to quantify the response rate and to generalise findings to all surgeons. Responses came mostly from urology and orthopaedics and therefore cannot be assumed to be representative of all disciplines: they reported a relatively low rate of video-assisted surgery, which may also suggest lack of generalisability. Despite efforts to disguise the study aims (through use of leisure-time physical activity-related questions on the first page of the online questionnaire), surgeons with musculoskeletal pain might have been more inclined to participate compared with others, potentially resulting in an overestimate of reported rates. However, we were able to collect a series of information about surgeons’ physical and psychological work environment, and their intention to retire. Our questionnaire assessed symptoms of pain over the past one month. We acknowledge that there may have been a lack of recall of symptoms that were more transient or minor but our focus was to understand the burden of musculoskeletal pain that might impact work, and not to be overly sensitive to all pain symptoms. This makes the survey unique in terms of exploring work environment, health, and retirement expectations of surgeons. Finally, it would have been of interest to collect data from patients treated by the surgeons to see if their perceptions about the performance of the surgeons were congruent, but that was not feasible within the scope of this study.

## 5. Conclusions

Overall, this study highlights the high physical and mental work demands on surgeons surrounding the pandemic. Such stressful environments, with predicted impact on personal lives, can affect musculoskeletal health and psychological wellbeing. Prolonged pain, especially at multiple anatomical sites, may compromise surgeons’ quality of life and their work performance, potentially resulting in earlier retirement. A follow-up study is recommended to assess whether demands have reduced. Additionally, we recommend that increasing awareness of surgeons’ work-related stresses and promotion of a supportive workplace environment are crucial for surgeons’ health and wellbeing and for patient care.

## Figures and Tables

**Figure 1 healthcare-13-00898-f001:**
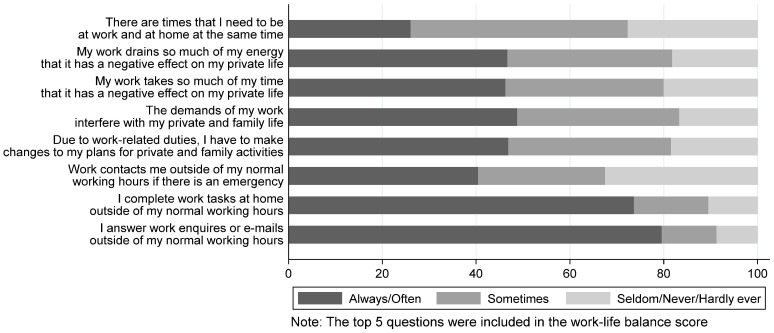
Self-reported work–life imbalance amongst 242 participating surgeons.

**Table 1 healthcare-13-00898-t001:** Characteristics and mental wellbeing of 242 participating surgeons.

	*N* (%)
Sex, Men	170 (70)
Age	
25–34	34 (14)
35–44	68 (28)
45–54	84 (35)
55–64	42 (17)
65+	14 (6)
Main surgical specialism	
Cardiothoracic	12 (5)
General surgery	21 (9)
Neurosurgery	5 (2)
Paediatric	8 (3)
Plastic	2 (1)
Trauma and orthopaedic	33 (14)
Urology	159 (66)
Academic	2 (1)
Leisure time physical activity (categories as per IPAQ categorisation scoring)	
Low	65 (27)
Moderate	55 (23)
High	120 (50)
Missing	2 (1)
Psychological wellbeing (WHO5)	55.8 (19)
Good mood	155 (64)
Low mood	85 (35)
Missing	2 (1)

**Table 2 healthcare-13-00898-t002:** Self-reported work characteristics of participating surgeons.

	*N* (%) Unless Otherwise Stated
Hours per week spent operating (median (IQR))	12 (8–15.5)
Operating time spent standing (median (IQR))	65 (35–100)
Operating time spent sitting (median (IQR))	35 (0–60)
Proportion (%) Operating (median (IQR))	
Open surgery	25 (10–80)
Thoracic surgery	0 (0–0)
Abdominopelvic surgery	20 (0–90)
Limb surgery	0 (0–0)
Imaging-guided surgery	10 (0–50)
Laparoscopic surgery	0 (0–25)
Robotic surgery	0 (0–10)
Occupational activities in a typical week	
Standing >3 h at a time	185 (76)
Working >2 h in total with your neck bent forward	176 (73)
Working >1/2 an hour in total with your neck twisted	150 (62)
Use of a keyboard >4 h in total	156 (64)
Other repeated movements of the wrist or fingers >4 h in total	150 (62)
Repeated bending and straightening of the elbow >1 h in total	117 (48)
Working >1 h with your hand or wrist twisted/rotated	137 (57)
Working >1 h in total with a tool that makes your hand(s) or arm(s) vibrate	29 (12)
Number of occupational activities in a typical week (median (IQR))	5 (3–6)
DCSQ job types	
Low strain (Low Demands/High Control)	57 (24)
Active job	30 (12)
Passive job	75 (31)
High strain (High Demands/Low Control)	71 (29)
Missing	9 (4)
Provision for physical comfort at work	
Yes	87 (36.0)
No	154 (63.6)
Missing	1 (0.4)
Working conditions	
Favourable	210 (87)
Unfavourable	24 (10)
Missing	8 (3)
Job dissatisfaction	
No	202 (83)
Yes	40 (17)

**Table 3 healthcare-13-00898-t003:** Prevalence rates and impact of regional musculoskeletal pain in participating surgeons.

Anatomical Site of Pain	Number (%) of Respondents Reporting Pain at This Anatomical Site	Burden from Musculoskeletal Pain at Each Anatomical Site Number (%) Who Reported Any of the Following: Interference with Activities, Healthcare Consultation or Medication Use for Pain at Each Anatomical Site			
		Interfered with Normal Activities (*N* (%) ^1^)	Interfered with Recreational Physical Activities (*N* (%) ^1^)	Led to Health Professional Advice/Treatment (*N* (%) ^1^)	Used Medication to Relieve Pain (*N* (%) ^1^)
Lower back	112 (46)	49 (44)	53 (47)	19 (17)	60 (54)
Neck	106 (44)	39 (37)	39 (37)	18 (17)	51 (48)
Shoulder	66 (27)	24 (36)	35 (53)	9 (14)	33 (50)
Hip	56 (23)	21 (38)	32 (57)	8 (14)	25 (45)
Wrist/Hand	45 (19)	12 (27)	13 (29)	7 (16)	17 (38)
Knee	39 (16)	13 (33)	22 (56)	3 (8)	17 (44)
Elbow	33 (14)	6 (18)	11 (33)	4 (12)	8 (24)
Ankle/Foot	29 (12)	13 (45)	20 (69)	5 (17)	17 (59)

^1^ % calculated over those reporting pain at the corresponding pain site.

**Table 4 healthcare-13-00898-t004:** Associations of musculoskeletal pain with demographic, personal and work-related factors in participating surgeons.

	Any Musculoskeletal Pain		Any Disabling ^1^ Musculoskeletal Pain	
	PRR	(95% CIs)	PRR	(95% CIs)
Sex (Female vs. Male)	1.08	(0.97, 1.20)	1.34	(0.97, 1.86)
Age				
35–44 (vs. 25–34)	0.97	(0.84, 1.12)	1.75	(1.02, 3.01)
45–54 (vs. 25–34)	0.94	(0.82, 1.09)	1.34	(0.75, 2.39)
55–64 (vs. 25–34)	0.89	(0.74, 1.07)	1.08	(0.55, 2.13)
65+ (vs. 25–34)	0.74	(0.49, 1.11)	0.8	(0.26, 2.52)
Psychological wellbeing (low vs. good mood)	1.03	(0.92, 1.16)	1.43	(1.05, 1.94)
Leisure time physical activity				
Moderate vs. Low	1.02	(0.87, 1.19)	0.7	(0.44, 1.10)
High vs. Low	1.02	(0.89, 1.17)	0.85	(0.60, 1.20)
Number of physical activities at work	1.04	(1.01, 1.07)	1.09	(1.00, 1.18)
DCSQ job type				
Active vs. Low strain	1.05	(0.88, 1.26)	0.86	(0.42, 1.77)
Passive vs. Low strain	0.96	(0.81, 1.13)	1.18	(0.73, 1.91)
High strain vs. Low strain	0.99	(0.84, 1.16)	1.55	(0.99, 2.42)
Work–life balance	0.92	(0.85, 0.99)	0.63	(0.51, 0.78)
Job dissatisfaction (Yes vs. No)	0.99	(0.85, 1.16)	1.29	(0.89, 1.88)
Working conditions (Unfavourable vs. Favourable)	1.06	(0.92, 1.22)	1.27	(0.80, 2.01)
Provision for physical comfort at work (No vs. Yes)	0.88	(0.79, 0.98)	0.84	(0.62, 1.15)

Note: All estimates shown in the table are sex- and age-adjusted. ^1^ Defined as pain that interfered with normal activities.

## Data Availability

The anonymised dataset analysed during the current study is available from the corresponding author on reasonable request.

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
