# Peer review of "Working Conditions, Musculoskeletal Pain and Wellbeing Among Hospital Surgeons: A Cross-Sectional Study"

_healthcare, 2025, doi:10.3390/healthcare13080898_

Round 1
Reviewer 1 Report
Comments and Suggestions for Authors
The study addresses a relevant aspect from the point of view of public health. The approach does not provide results that can be immediately transferred to the general population, but it represents a way of presenting the reality of the health service that deserves attention and interest from health decision makers.
The descriptive statistics are adequate for the purpose of the study.
The limitations are clearly expressed, and the conclusions are consistent with the data presented. There are some small suggestions to improve the readability of the text.
Table 3 is very complex and could be improved.
Table 4 : It seems that with increasing age the association of work-related risk factors and musculoskeletal pain decreases
Lines 238, 246 : Our study demonstrated that surgeons were experiencing work-life conflict, evident in frequent work-related tasks outside of working hours, and despite moderate levels of overall job dissatisfaction. Lines 246 and following: This aspect is interesting and should be highlighted.
Author Response
The study addresses a relevant aspect from the point of view of public health. The approach does not provide results that can be immediately transferred to the general population, but it represents a way of presenting the reality of the health service that deserves attention and interest from health decision makers.
Response: Thank you for your positive feedback.
The descriptive statistics are adequate for the purpose of the study.
The limitations are clearly expressed, and the conclusions are consistent with the data presented. There are some small suggestions to improve the readability of the text.
Response: Thank you
Table 3 is very complex and could be improved.
Response: Thank you for this comment. We have re-labelled Table 3 to clarify that the 3rd-6th columns relate to how burdensome the pain at each anatomical site was, as reported by individuals with pain at that site. To make the table easier to digest we have removed absolute N. and only kept percentages throughout, as below:
Anatomical site of pain |
Number (%) of respondents reporting pain at this anatomical site |
Burden from musculosketal pain at each anatomical site Number (%) who reported any of: intereference with activities, healthcare consultation or medication use for pain at each anatomical site |
|||
|
|
Interfered with normal activities (N (%)1) |
Interfered with recreational physical activities (N (%)1) |
Lead to health professional advice/treatment (N (%)1) |
Used medication to relieve pain (N (%)1) |
Low back |
112 (46) |
49 (44) |
53 (47) |
19 (17) |
60 (54) |
Neck |
106 (44) |
39 (37) |
39 (37) |
18 (17) |
51 (48) |
Shoulder |
66 (27) |
24 (36) |
35 (53) |
9 (14) |
33 (50) |
Hip |
56 (23) |
21 (38) |
32 (57) |
8 (14) |
25 (45) |
Wrist/Hand |
45 (19) |
12 (27) |
13 (29) |
7 (16) |
17 (38) |
Knee |
39 (16) |
13 (33) |
22 (56) |
3 (8) |
17 (44) |
Elbow |
33 (14) |
6 (18) |
11 (33) |
4 (12) |
8 (24) |
Ankle/Foot |
29 (12) |
13 (45) |
20 (69) |
5 (17) |
17 (59) |
Table 4 : It seems that with increasing age the association of work-related risk factors and musculoskeletal pain decreases
Thank you for your comment. Table 4 shows associations between each of the demographic, personal and work-related factors and risk of reporting any musculoskeletal pain (and disabling musculoskeletal pain) after adjustment for age and sex. Therefore, the results show that there is no association between age and reporting any musculoskeletal pain (each of the 95% confidence intervals embrace 1.0). However, younger surgeons appear more likely than older ones to report any disabling musculoskeletal pain.
Lines 238, 246 : Our study demonstrated that surgeons were experiencing work-life conflict, evident in frequent work-related tasks outside of working hours, and despite moderate levels of overall job dissatisfaction. Lines 246 and following: This aspect is interesting and should be highlighted.
Response: Thank you – we added the work importantly” to that sentence, as follows:
“Our study importantly demonstrated that surgeons were experiencing work-life conflict, evident in frequent work-related tasks outside of working hours, often disrupting private/family plans.”
Reviewer 2 Report
Comments and Suggestions for Authors
This study is interesting and addresses a relevant topic. The study explored the biomechanical and psychosocial challenges faced by hospital surgeons in the UK post-pandemic. It focused on their experience with pain, work-life balance, and job satisfaction. An online survey, based on validated tools, was distributed to surgeons. The results showed that surgeons often reported strenuous work, frequent musculoskeletal pain, and poor work-life balance. Pain was more prevalent among those with physically demanding work, while better work-life balance had a protective effect. Job dissatisfaction was linked to the intention of early retirement. The study highlighted the physical and emotional toll on surgeons, potentially affecting their job satisfaction and career longevity, suggesting a need for further research.
Commentaries :
L86-90: The data collection presents a risk of recall bias, as surgeons may forget the period during which they experienced pain. How did the authors minimize this risk?
L298-309: One limitation of this study, not mentioned by the authors, is the absence of patient perceptions regarding the performance of the surgeons.
Author Response
This study is interesting and addresses a relevant topic. The study explored the biomechanical and psychosocial challenges faced by hospital surgeons in the UK post-pandemic. It focused on their experience with pain, work-life balance, and job satisfaction. An online survey, based on validated tools, was distributed to surgeons. The results showed that surgeons often reported strenuous work, frequent musculoskeletal pain, and poor work-life balance. Pain was more prevalent among those with physically demanding work, while better work-life balance had a protective effect. Job dissatisfaction was linked to the intention of early retirement. The study highlighted the physical and emotional toll on surgeons, potentially affecting their job satisfaction and career longevity, suggesting a need for further research.
Thank you for your careful review of our manuscript.
Commentaries :
L86-90: The data collection presents a risk of recall bias, as surgeons may forget the period during which they experienced pain. How did the authors minimize this risk?
Response: To minimise the risk of recall bias, the questionnaire asked surgeons to report on musculoskeletal pain experienced within the past one month. The Reviewer is correct that surgeons may have forgotten more trivial or transient symptoms but our particular focus was on symptoms causing disability or healthcare utilisation, which we assessed in Table 3, as these are the types of symptoms likely to have some impact on the work surgeons do. We have added this comment in Limitations, lines 310-314:
“Our questionnaire assessed symptoms of pain over the past one month. We acknowledge that there may have been a lack of recall of symptoms that were more transient or minor but our focus was to understand the burden of musculoskeletal pain that might impact work and not to be overly sensitive to all pain symptoms”.
L298-309: One limitation of this study, not mentioned by the authors, is the absence of patient perceptions regarding the performance of the surgeons.
Response: Thank you for your comment. We have added this as a possible limitation of the study, as follows, line 315-318:
“Finally, it would have been of interest to collect data from patients treated by the surgeons to see if their perceptions about the performance of the surgeons were congruent but that was not feasible within the scope of this study”.
Reviewer 3 Report
Comments and Suggestions for Authors
Dear authors, thank you for the opportunity to get acquainted with your study.
The study is devoted to the study of well-being, health of the musculoskeletal system and working conditions and psychosocial factors in surgeons.
The authors conducted a study on a representative sample using validated questionnaires, applying statistical methods. The authors presented the results in a structured manner, illustrated them with tables and a figure.
At the same time, there are a number of comments on the manuscript that require elimination for a better understanding of the study:
1. The main question that requires resolution: the goal and hypotheses. They are not indicated by the authors, because of which it is unclear what the authors wanted to clarify and how the data analysis is further constructed. This needs to be eliminated.
2. The title of the manuscript is very broad, the study is more specific. It is necessary to adjust the title of the manuscript closer to the goal.
3. The introduction needs to expand the data, what has already been studied regarding surgeons? what aspects of their health and psychosocial factors have been studied the most? what has been established? What scientific gap in knowledge do the authors fill with their research?
4. It is necessary to justify the choice of factors that were included in the analysis. In this regard, why were these particular factors selected from the many?
5. Section 2.3. It is necessary to structure and justify the choice of questions. If the authors used modifications, then provide links. Is Cronbach's alpha conducted to check the consistency of the blocks?
6. Specify the sample of doctors: country, regions where the surveys were conducted.
7. In this regard, there is no information on length of service in the position? This could have affected the results.
8. Prescribe statistical methods for specific hypotheses. Specify which statistical method and what criteria did the authors use?
9. Table 4 is not entirely clear. Was this compiled based on the results of the application of what methods? Was it compiled to test several hypotheses at once? It would be more convenient to see a separate table for each hypothesis or to present and describe it more clearly in relation to Table 4. Because the relationship of factors with possible leaving work is not obvious from the table. 10. The discussion of the results should be changed according to the tested hypotheses. Also, what practical recommendations can be given based on the results of the study?
11. The conclusions should be changed to be clearer, according to the tested hypotheses.
In connection with the above, the article requires revision.
Best wishes and respect for your work, reviewer
Author Response
Dear authors, thank you for the opportunity to get acquainted with your study.
The study is devoted to the study of well-being, health of the musculoskeletal system and working conditions and psychosocial factors in surgeons.
The authors conducted a study on a representative sample using validated questionnaires, applying statistical methods. The authors presented the results in a structured manner, illustrated them with tables and a figure.
At the same time, there are a number of comments on the manuscript that require elimination for a better understanding of the study:
1. The main question that requires resolution: the goal and hypotheses. They are not indicated by the authors, because of which it is unclear what the authors wanted to clarify and how the data analysis is further constructed. This needs to be eliminated.
Response:The penultimate paragraph of Introduction has now been re-phrased as the aim, as follows:
“Thus, the aim of this study was to obtain a post-pandemic understanding of surgeons’ working conditions, psychological wellbeing and musculoskeletal pain in order to facilitate workforce planning and maximise the efficiency of surgical practice while fostering a healthier and more resilient healthcare workforce”.
- The title of the manuscript is very broad, the study is more specific. It is necessary to adjust the title of the manuscript closer to the goal.
Response: The title has been altered as suggested, and now is:
“Working Conditions, Musculoskeletal pain and Wellbeing Among Hospital Surgeons: A Cross-Sectional Study”
The introduction needs to expand the data, what has already been studied regarding surgeons? what aspects of their health and psychosocial factors have been studied the most? what has been established? What scientific gap in knowledge do the authors fill with their research?
Response. Our aim in the Introduction was to be brief. However, we have expanded our previous information about the musculoskeletal symptoms previously documented in surgeons, including adding some additional references /systematic reviews, lines 63-76:
“Surgeons were at the forefront of restoring patient care and addressing the increased waiting-lists. Amongst working-aged adults, the leading causes of sickness absence, presenteeism and work disability are musculoskeletal disorders and common mental health conditions (9). Globally, healthcare struggles to retain its workforce, particularly doctors and nurses (10). From 2007 to 2020, the number of doctors working in the NHS opting for early retirement trebled (11), impacting patient care, health service delivery and surgical training (12). Even pre-pandemic, there was evidence that surgeons were commonly reporting musculoskeletal disorders (13-16) with conflicting evidence as to whether pain was increased or decreased by robotic /video-assisted techniques (14,15). Thus, the aim of this study was to obtain a post-pandemic understanding of surgeons’ working conditions, perceived work-life conflict, psychological wellbeing, retirement intentions and experiences of musculoskeletal pain in order to facilitate workforce planning and maximise the efficiency of surgical practice while fostering a healthier and more resilient healthcare workforce”.
- It is necessary to justify the choice of factors that were included in the analysis. In this regard, why were these particular factors selected from the many?
Response: Our aim was to develop an evidence-informed short questionnaire that could be readily completed on-line by busy hospital surgeons. Specific measures were chosen for each of: psychological wellbeing; ergonomic exposures; psychosocial exposures; work-life conflict; job satisfaction and working conditions. A paragraph has been added as follows, lines 102-104:
“Relevant covariates were chosen to be brief and cover each of the following specific domains: psychological wellbeing; ergonomic exposures; psychosocial exposures; work-life conflict; job satisfaction and working conditions”.
Section 2.3. It is necessary to structure and justify the choice of questions. If the authors used modifications, then provide links. Is Cronbach's alpha conducted to check the consistency of the blocks?
Response: We did not modify any questionnaires ourselves. We used the validated modified version of the Karasek questionnaire and cited the relevant citation:
“Work-related demand, control and support was assessed using questions from the validated Swedish Demand-Control-Support Questionnaire (DCSQ) (18), the modified version of Karasek’s questionnaire (19)”.
Specify the sample of doctors: country, regions where the surveys were conducted.
Resposne: snowball sampling was used so that the questionnaire could ahev been answerd by any qualified surgeon but the active circulation was predominatly in UK and USA. This has been added, as follows”
“A brief questionnaire entitled “The health, work and wellbeing of Surgeons” was developed on the Qualtrics© on-line platform. The survey was publicised through co-authors’ surgical networks predominantly in the UK but also in USA using societies and social media and snowball sampling between Oct 2021 and Jan 2022”.
In this regard, there is no information on length of service in the position? This could have affected the results.
Response: Thank you. Yes, we collected information about years since qualification to explore this risk factor but found no effect of this variable and therefore did not present these data.
Prescribe statistical methods for specific hypotheses. Specify which statistical method and what criteria did the authors use?
Response: Thank you for the opportunity to clarify. This was an exploratory study aimed at describing working conditions, perceived work-life conflict, experiences of musculoskeletal pain and thoughts about retirement in a sample of contemporary surgeons. As such, we did not test specific hypotheses but instead described the characteristics of a sample of surgeons. As specified in section 2.4, we employed counts and percentages for descriptive analyses. When exploring the association between a panel of risk factors and self-reported musculoskeletal pain (and disabling musculoskeletal pain) we used Poisson regression with robust standard errors. Similarly, we used Poisson regression with robust standard errors to explore risk factors for intending to retire before reaching state pension age. We have now expanded this section to specify that mutually adjusted models, including factors significant when models were only adjusted for sex and age, were also performed:
“We used counts and percentages to describe the study population, their characteristics, and musculoskeletal pain. We used Poisson regression models with robust standard errors to explore the effects of personal, demographic factors, work-related physical and mental influences on musculoskeletal pain, and those on intention to retire before state pension age. The effects were summarized by prevalence rate ratios (PRRs) and 95% confidence intervals (CIs) and were sex- and age-adjusted. A final mutually adjusted model was also computed with estimates adjusted for age, sex and all factors significant in the minimally adjusted models.
Data were analysed using Stata v17.0 software”.
Table 4 is not entirely clear. Was this compiled based on the results of the application of what methods? Was it compiled to test several hypotheses at once? It would be more convenient to see a separate table for each hypothesis or to present and describe it more clearly in relation to Table 4. Because the relationship of factors with possible leaving work is not obvious from the table.
We apologise for the lack of clarity of Table 4. This table shows associations between a panel of risk factors (first column) and reporting any musculoskeletal pain (second column) or any musculoskeletal pain which was affecting activities (defined as “disabling”). Each estimate expressed as a prevalence rate ration (PRR) and 95% confidence interval (95%CI) originates from a separate model adjusted for age and sex (as mentioned in the table footnote). To clarify, we have now specified that mutually adjusted models reported in the text are not reported in the table. Additionally, the associations between risk factors and intention to retire early are also not reported in the table and this is now clarified.
- The discussion of the results should be changed according to the tested hypotheses. Also, what practical recommendations can be given based on the results of the study?
Response: as pointed out above, this was an observational epidemiological study designed to explore the burden of physical and psychological symptoms on contemporary surgeons in the aftermath of the COVID-19 pandemic. The Discussion explains these findings in context with thre results from other studies.
The conclusions should be changed to be clearer, according to the tested hypotheses.
In connection with the above, the article requires revision. Best wishes and respect for your work, reviewer
Response: We have re-worded the Conclusions to make it clear that our text is a recommendation:
“Overall, this study highlights the high physical and mental work demands on surgeons surrounding the pandemic. Such stressful environments, with predicted impact on personal lives, can affect musculoskeletal health and psychological wellbeing. Prolonged pain, especially at multiple anatomical sites, may compromise both surgeons’ quality of life and their work performance, potentially resulting in earlier retirement. A follow-up study is recommended to assess whether demands have reduced. Additionally, we recommend that increasing awareness of surgeons’ work-related stresses and promotion of a supportive workplace environment are crucial for surgeons’ health and wellbeing and for patient care.”
Round 2
Reviewer 3 Report
Comments and Suggestions for Authors
Dear authors, thank you for your clarifications and additions.
The article can be recommended for publication.
Best wishes, reviewer